

# The right thalamus may play an important role in anesthesia-awakening regulation in frogs

Yanzhu Fan[1,2], Xizi Yue[1,3], Fei Xue[1], Steven E. Brauth[4], Yezhong Tang[1] and Guangzhan Fang[1,3]

[1] Chengdu Institute of Biology, Chinese Academy of Sciences, Chengdu, Sichuan, People's Republic of China
[2] University of Chinese Academy of Sciences, Beijing, People's Republic of China
[3] College of Life Sciences, China West Normal University, Nanchong, Sichuan, People's Republic of China
[4] Department of Psychology, University of Maryland, United States of America

## ABSTRACT

**Background**. Previous studies have shown that the mammalian thalamus is a key structure for anesthesia-induced unconsciousness and anesthesia-awakening regulation. However, both the dynamic characteristics and probable lateralization of thalamic functioning during anesthesia-awakening regulation are not fully understood, and little is known of the evolutionary basis of the role of the thalamus in anesthesia-awakening regulation.

**Methods**. An amphibian species, the South African clawed frog (*Xenopus laevis*) was used in the present study. The frogs were immersed in triciane methanesulfonate (MS-222) for general anesthesia. Electroencephalogram (EEG) signals were recorded continuously from both sides of the telencephalon, diencephalon (thalamus) and mesencephalon during the pre-anesthesia stage, administration stage, recovery stage and post-anesthesia stage. EEG data was analyzed including calculation of approximate entropy (ApEn) and permutation entropy (PE).

**Results**. Both ApEn and PE values differed significantly between anesthesia stages, with the highest values occurring during the awakening period and the lowest values during the anesthesia period. There was a significant correlation between the stage durations and ApEn or PE values during anesthesia-awakening cycle primarily for the right diencephalon (right thalamus). ApEn and PE values for females were significantly higher than those for males.

**Discussion**. ApEn and PE measurements are suitable for estimating depth of anesthesia and complexity of amphibian brain activity. The right thalamus appears physiologically positioned to play an important role in anesthesia-awakening regulation in frogs indicating an early evolutionary origin of the role of the thalamus in arousal and consciousness in land vertebrates. Sex differences exist in the neural regulation of general anesthesia in frogs.

Corresponding author
Guangzhan Fang, fanggz@cib.ac.cn

## INTRODUCTION

General anesthesia (GA) is a drug-induced depression of the central nervous system (CNS) that permits long-term operations and experimental studies requiring invasive procedures (*Antognini, Barter & Carstens, 2005*; *Goddard & Smith, 2013*), which is characterized by unawareness or unconsciousness, analgesia and immobilization (*Brown, Lydic & Schiff, 2010*; *Franks, 2008*; *Pocock & Richards, 1993*). Previous studies have shown that specific brain regions, including the midbrain reticular formation, thalamus and some brain regions located within the parietal and frontal association cortex, are more sensitive to anesthesia than others (*Heinke & Koelsch, 2005*). Furthermore functional-brain-imaging data in humans reveal suppression of activity in specific brain areas including thalamic nuclei and the midbrain reticular formation during general anesthesia (*Alkire, Haier & Fallon, 2000*). Anesthetics can suppress cortico-thalamo-cortical activity and produce unconsciousness by reducing membrane excitability in the thalamo-cortical loop, which has been proposed to play a crucial role in the anesthesia-awakening cycle (*Ries & Puil, 1999*). In particular, many neurons within the thalamus exhibit anatomical and physiological specializations which support large-scale cerebral dynamics related to consciousness (*Domino, 1968*). Thus the thalamus, as an information hub, seems well positioned to function as a key region for regulation of anesthesia-induced unconsciousness by gating sensory information processing (*Franks, 2008*; *Ries & Puil, 1999*). Nevertheless, the dynamic properties of the thalamus during the anesthesia-awakening cycle remain unclear.

Lateralization or asymmetry of cerebral function, has been described in many vertebrate and invertebrate taxa (*Lippolis et al., 2002*; *Rogers, 2014*; *Rogers et al., 2013*; *Rogers & Vallortigara, 2015*; *Rogers & Vallortigara, 2017*; *Rosa-Salva et al., 2012*; *Roussigné, Blader & Wilson, 2012*; *Vallortigara et al., 1998*), and appears to be a fundamental aspect of nervous system organization. Brain lateralization may enable simultaneous channeling of different types of information into lateralized brain circuits thereby enabling separate and parallel processing in the two hemispheres (*Dadda et al., 2009*; *Fang et al., 2014*; *Rogers & Vallortigara, 2015*; *Rogers, Vallortigara & Andrew, 2013*; *Vallortigara & Rogers, 2005*). Relatively few neurophysiological studies have focused on whether regulation of general anesthesia is also asymmetric. The purpose of this paper is to investigate the neural mechanisms of general anesthesia in an amphibian species, the South African clawed frog (*Xenopus laevis*), an important animal model for developmental and genetic studies (*Schultz & Dawson, 2003*), which has been frequently used for basic vertebrate nervous system functioning (*Guénette, Giroux & Vachon, 2013*), to test the theory that lateralized thalamic general anesthesia regulation is an evolutionarily conserved feature of land vertebrates.

The electroencephalogram (EEG) reflects the summed post-synaptic potentials generated by pyramidal cells of the cerebral cortex and can be recorded on the surface of the scalp (*Muthuswamy, Roy & Sharma, 1996*). Though the EEG allows assessing brain activity during different anesthesia stages, results of traditional linear analysis based on raw EEG data are difficult to interpret precisely (*Billard et al., 1997*; *Bruhn et al., 2006*; *Katoh, Suzuki & Ikeda, 1998*; *Mahon et al., 2008*). This is because neuronal ensembles exhibit important
nonlinear behaviors whose characteristics can only be adequately described using nonlinear parameters (*Bruhn, Röpcke & Hoeft, 2000*; *Burioka et al., 2005*). Among these methods, both approximate entropy (ApEn) and permutation entropy (PE) are statistical parameters that can quantify randomness and the predictability of a time series and can be used to depict the complexity of EEG signals and the effects of anesthetic drugs on the CNS (*Bruhn, Röpcke & Hoeft, 2000*; *Fan et al., 2011*; *Li, Cui & Voss, 2008*; *Liang et al., 2015*; *Ouyang et al., 2010*; *Pincus, 1991*).

The amphibian brain exhibits the same segmental architecture as birds, reptiles and mammals including humans and is composed of the telencephalon, diencephalon, mesencephalon, metencephalon and myelencephalon (*Wilczynski & Endepols, 2007*). The diencephalon consists of two main cellular aggregates, the thalamus dorsally and the hypothalamus ventrally, which exhibit the same general patterns of connectivity as amniotes (*Butler, 1995*; *Laberge et al., 2008*). Anatomically the diencephalon is caudal to the telencephalon and rostral to the mesencephalon and located below the skull at a point where the cerebrum is disappearing. Thus, it is possible to implant electrodes in the diencephalon and to obtain high signal-to-noise ratio EEG activity originating in the thalamus.

Triciane methane sulfonate (MS-222), which can block motor activity and nociception through relatively long term blockade of action potential initiation via voltage gated sodium channels in the brain and muscles (*Ramlochansingh et al., 2014*), is widely used to induce anesthesia for amphibians (*Downes, 1995*; *Lalonde-Robert, Beaudry & Vachon, 2012*). In addition, a previous EEG study demonstrated that MS-222 can lead to profound CNS depression and is capable of causing unconsciousness in *X. laevis* (*Lalonde-Robert et al., 2012*). For example, *X. laevis* can be anesthetized effectively with concentrations ranging from 1 to 5g/L MS-222 solution (*Torreilles, McClure & Green, 2009*).

This study was conducted on *X. laevis* frogs with implanted electrodes in the telencephalon, diencephalon and mesencephalon, respectively. The animals were general anesthetized by immersion in MS-222. EEG signals were obtained during the pre-anesthesia, administration, recovery and post-anesthesia stages and were recorded continuously. The ApEn and PE for each stage were calculated in order to explore the relationships between neural activities recorded in each brain area across the anesthesia-awakening cycle including possible functional lateralization for general anesthesia regulation.

## MATERIAL AND METHODS

### Animal

Fourteen South African clawed frogs of both sexes (seven males and seven females) bred in the lab were used in the present study. The subjects were separated by sex and raised in two aquaria (120 × 50 cm and 60 cm deep) with water depth approximately 20 cm. The animals were fed every three days and the water was replaced once a week. The aquaria were placed in a room in which the temperature was maintained at 20 ± 1 °C with a 12/12 light-dark cycle (lights on at 08:00 h). The subjects measured 8.1 ± 1.1 cm (mean ± SD) in body length and 67.1 ± 22.2 g in body mass at the time of surgery. All surgery was performed

under MS-222 anesthetic, and all efforts were made to minimize discomfort. All animal procedures were carried out in accordance with the Animal Care and Use Committee of Chengdu Institute of Biology, Chinese Academy of Sciences (Approval number: 2016005).

## Surgery

All experiments were conducted during April to May (this species breeds between April and September in our lab), 2016. The animals were deeply anesthetized via water bath using 500 ml of MS-222 solution (3.5 g/L) and buffered by adding sodium bicarbonate to achieve a neutral pH (7.0–7.4). The optimum depth of anesthesia for surgery was determined to be when the withdrawal reflex to toe pinching (i.e., the toe pinch response by grasping a digit with the tweezers) is lost, i.e., responsiveness to pinch limb test is used as a proxy for testing consciousness status. MS-222 solution was wiped to the animal's skin using a cotton swab when it was necessary during surgery. The animals were wrapped with wet cotton gauze. Seven cortical EEG electrodes, composed of miniature stainless steel screws ($\varphi$ 0.8 mm), were inserted in the skull by turning 3.5 rotations to implant at a depth of about 1.1 mm: the left and right sides of the telencephalon, diencephalon and mesencephalon (LT, RT, LD, RD, LM and RM) and referenced to the electrode above the cerebellum (P) (Fig. 1). Ten seconds of typical EEG waves are presented along with the corresponding electrode pairs in Fig. 1. The electrodes above LT and RT were implanted bilaterally 6.4 mm anterior to the lambda (i.e., the vertex where the skull sutures intersect) and 1.0 mm lateral to the midline respectively, and the electrodes above LD and RD were implanted bilaterally 3.4 mm anterior to the lambda and 1.0 mm lateral to the midline respectively, while the electrodes above LM and RM were implanted bilaterally 1.4 mm anterior to the lambda and 1.0 mm lateral to the midline, respectively. P was implanted 1.0 mm posterior to the lambda at the midline (Fig. 1). One end of all electrode leads, formvar-insulated nichrome wires, was twined tightly on the screws and fixed on the skull of the frog with dental acrylic, while the other end was soldered to the pins of the light connector. Finally, the skin edges and muscles surrounding the wound were treated with the ointment with triple antibiotic and pain relief (CVS pharmacy, Woonsocket, RI, USA) to prevent infection and discomfort.

Each frog was housed individually for one day for recovery before the following experiments were performed. After the end of all experiments, the subjects were euthanized by immersion in MS-222 solution for a prolonged period of time and the electrode locations were confirmed by injecting hematoxylin dye through the skull holes in which the electrodes had previously been installed (Fig. S1).

## Data acquisition

The experiments were performed in a soundproof and electromagnetically shielded chamber in which the background noise was 24.3 $\pm$ 0.7 dB (mean $\pm$ SD) within a transparent plastic box with a floor area of 18 $\times$ 11 cm$^2$, an upper cover area of 20 $\times$ 13 cm$^2$ and 12 cm in height. A sponge (17 $\times$10$\times$ 1 cm) which had absorbed about 200 ml of water was placed at the bottom of the box. Lights and temperature in the chamber were maintained as in the housing room. A video camera with an infrared light source
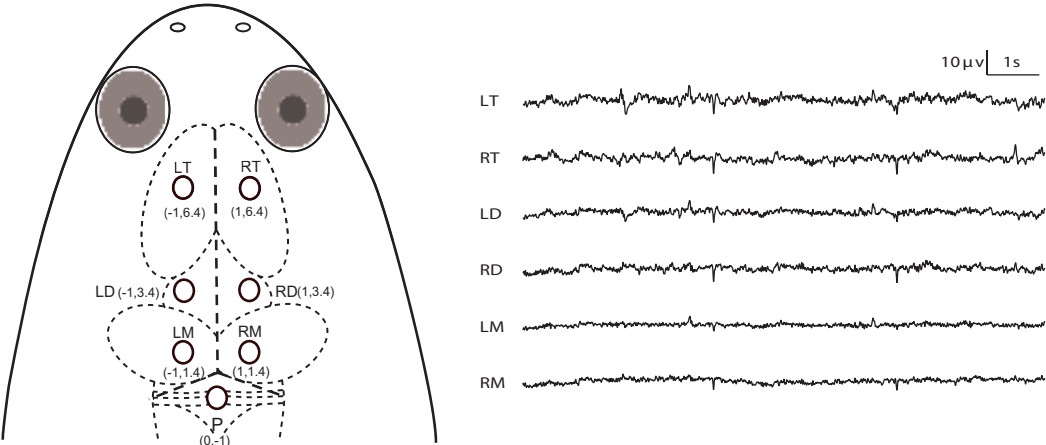

**Figure 1 Electrode placements and 10 s of typical artifact-free EEG tracings for each channel during Stage I.** The intersection of the three dashed lines in the head of *X. laevis* denotes the lambda (i.e., the vertex where the skull sutures intersect). Abbreviations: LT and RT, the left and right telencephalon; LD and RD, the left and right diencephalon; LM and RM, the left and right mesencephalon.

and motion detector was appended centrally approximately 40 cm above the box for monitoring the subject's behavior from outside the chamber.

Before the experiments began the subject was connected to the signal acquisition system (RM6280C; Chengyi, Sichuan, China). Both EEG and behavioral data were recorded during four stages: (1) During Stage I, the awake animal was placed in the experimental box and free to move for 30 min before anesthesia; in fact the subject usually kept immobility with its head towards one corner of the box; (2) during Stage II, the subject was transferred to another box (of the same size within the experimental chamber) containing a 200 ml MS-222 solution (3.5 g/L) with a depth about 1 cm, in which the subject was immersed for about 5 min until the toe pinch response (grasping one of the four frog's digits selected randomly every 30 s) disappeared, and then subsequently removed quickly from the MS-222 solution; (3) Stage III was defined as the time period between returning the subject to the experimental box and the moment locomotor activity first reappeared while one of the digits was selected randomly to touch every 30 s by a cotton swab; (4) during Stage IV the recovered animal was allowed to produce voluntary movements for 30 min after anesthesia.

## Approximate entropy (ApEn)

ApEn is a measure of irregularity or complexity of a dynamical system as proposed by Pincus (*Pincus, 1991*), which is particularly effective for analyzing short and noisy time-series data and which can categorize a wide variety of systems ranging from multi periodic, stochastic to mixed systems (*Pincus, 1995a*; *Pincus, 1995b*). Since ApEn is a basically model-independent regularity statistic unrelated to signal magnitude and can be applied to relatively noisy and non-stationary physiological time series of short length (*Pincus, 1991*; *Pincus, 1995a*; *Pincus, 1995b*), it is one of the most widely used nonlinear methods in the field of anesthesia and EEG (*Bruhn, Bouillon & Shafer, 2001*; *Bruhn, Röpcke*

& *Hoeft, 2000*; *Bruhn et al., 2000*; *Fan et al., 2011*; *Hudetz, 2002*; *Hudetz, Wood & Kampine, 2003*; *Koskinen et al., 2006*; *Kreuzer et al., 2010*; *Noh et al., 2006*; *Sleigh et al., 2004*). The procedure for estimating ApEn is described as follows.

For a given time series $u(i), i = 1, \ldots, N$ from measurements equally in time, form a sequence of vectors that are defined by

$$X(i) = [u(i), u(i+1), \ldots, u(i+m-1)], i = 1, \ldots, N - m + 1 \tag{1}$$

where $m$ is the embedding dimension of phase space. The distance between vectors $X(i)$ and $X(j)$ can be defined as

$$d[X(i), X(j)] = \max_{k=1,2,\ldots,m} |u(i+k-1) - u(j+k-1)|. \tag{2}$$

For a given $i \leq N - m + 1$, $N^m(i)$ is the number of $j$ in dimension $m$ such that $d[X(i), X(j)] \leq r$, then $C_i^m(r)$ is defined as

$$C_i^m(r) = (N - m + 1)^{-1} N^m(i) \tag{3}$$

where $r$ is the tolerance (i.e., previous setting of minimal distance between vectors $X(i)$ and $X(j)$). Next step is to compute the natural logarithm of each $C_i^m(r)$ and average it over $i$

$$\varnothing^m(r) = (N - m + 1)^{-1} \sum_{i=1}^{N-m+1} \ln C_i^m(r). \tag{4}$$

Then increase the embedding dimension, i.e., from $m$ to $m+1$. Repeat steps (3)–(4) to obtain $C_i^{m+1}(r)$ and $\phi^{m+1}(r)$.

Finally, ApEn can be calculated as

$$ApEn(m, r, N) = \phi^m(r) - \phi^{m+1}(r). \tag{5}$$

Mathematically, ApEn measures the likelihood that runs of patterns which are close for $m$ observations will remain close on the following incremental comparisons (*Jaušovec & Jaušovec, 2010*). Thus, ApEn is a non-negative number calculated from a time series using the above protocols. Smaller values of ApEn imply a stronger regularity or persistence in the time series while larger values indicate greater fluctuation or irregularity. Usually the parameter $m$ is set to 1 or 2 ($m = 2$ recommended) while $r$ can range from 0.1 to 0.25 times the SD of the original data sequence (*Pincus, 1991*).

## Permutation entropy (PE)

Bandt and Pompe recently proposed a new permutation method to map a continuous time series onto a symbolic sequence, where the statistics of the symbolic sequences are called permutation entropy (PE) (*Bandt & Pompe, 2002*). PE is an appropriate complexity measure for chaotic time series, in particular in the presence of dynamical and observational noise. The advantages of PE are its simplicity, extremely fast calculation, its robustness and invariance with respect to non-linear monotonous transformations (*Bandt & Pompe, 2002*).

Given a time series $x_t$, $(t = 1, 2, 3 \ldots)$, an embedding procedure forms vectors $X_t[x_t, x_{t+\tau}, \ldots x_{t+m\tau}]$ with the embedding dimension $m$ and the lag $\tau$. The vector $X_t$

can be arranged in an increasing order. For different $m$, there will be $m!$ possible order patterns, which are also called permutations. Considering each permutation as a symbol, the vectors $X_t$ can be represented by a symbol sequence; the distinct number of symbols ($J$) should be less than or equal to $m!$, namely $J \leq m!$ For the time series $x_t$, the probability distributions of the distinct symbols are defined as $p_1, p_2, \ldots p_j$; the PE of this time series is defined by

$$H_p(m) = -\sum_{j=1}^{J} p_j \ln p_j. \qquad (6)$$

The corresponding normalized entropy can be defined as follows:

$$H_p = H_p(m)/\ln(m!). \qquad (7)$$

The largest value of $H_p$ is one, meaning the time series is completely random; while the smallest value of $H_p$ is zero, meaning the time series is very regular. In short, the permutation entropy refers to the local order structure of the time series, which can give a quantitative complexity measure for a dynamical time series.

PE calculation depends on the selection of time interval $N$ and embedding dimension $m$. When $m$ is too small (less than 3), the scheme will not work well since there are only a few distinct states for EEG recordings. For practical purposes, $m = 3, \ldots, 7$ is recommended (*Bandt & Pompe, 2002*), and that for a long EEG recording, a large value of $m$ is better (*Li, Cui & Voss, 2008*). On the other hand, the length of the EEG recording should be larger than $m!$ in order to achieve a proper differentiation between deterministic and stochastic dynamics (*Li et al., 2014*; *Ouyang et al., 2010*). In addition, to allow every possible order pattern of dimension $m$ to occur in a time series of length $N$, the condition $m! \leq N - (m-1)\tau$ must hold. Moreover, $N \gg m! + (m-1)\tau$ is required to avoid undersampling (*Amigó, Zambrano & Sanjuán, 2007*). For this reason, given $m$ dimensions, we need to choose $N \gg (m+1)!$. Since this study concentrates on the detection of dynamical changes in the EEG recording, too large a value of $m$ or $N$ would be inappropriate. Therefore $m = 5$ and $N = 1,000$ were used for calculating PE in the light of previous studies (*Li, Cui & Voss, 2008*; *Liang et al., 2015*; *Ouyang et al., 2010*).

## Data processing

In order to evaluate the appropriate parameters for ApEn, five minutes of EEG data during Stage I were selected randomly. After band-pass filtering (0.5–45 Hz) and downsampling at 256 Hz, $r$ and $N$ were determined by calculating ApEn ($m$, $r$, $N$) with increasing $r$ from 0.1 to 0.4 SD in steps of 0.05 and $N$ from 100 to 2,000 in steps of 100 for a few randomly selected EEG segments while $m = 2$. ApEn reached its maximum on a plateau when $N = 500$ and this plateau was stable only when $r = 0.15$ SD. Therefore, in the present study, ApEn for EEG data was computed using a slide window of $N = 512$ (2 s of the EEG signal) overlapping $N/2$ for each step while $r = 0.15$ SD, to yield an ApEn vector. Furthermore, in accordance with previous studies (*Li, Cui & Voss, 2008*; *Liang et al., 2015*; *Ouyang et al., 2010*), each epoch with $N = 1,000$ (i.e., 10 s of the EEG signal) overlapping

$N/2$ was used after band-pass filtering (0.5–45 Hz) and downsampling at 100 Hz for calculating PE.

Any epoch with an amplitude extremum beyond $\pm 100\ \mu v$ was discarded as artifact. The designation of artifact in one channel resulted in removal of data in all other channels in order to ensure that datasets derived from all channels were derived from the same time periods. ApEn or PE was then averaged for each stage, each channel and each subject based on artifact-free epochs.

## Statistical analyses

The normality of the distribution and homogeneity of variance for ApEn and PE values were estimated with the Shapiro–Wilk $W$ test and Levene's test, respectively. ApEn and PE values were statistically analyzed using a three-way repeated measured ANOVA with the variables of "sex" (male/female), "stage" (Stages I, III and IV, Stage II was not included because of too many artifacts), and "brain area" (LT, RT, LD, RD, LM and RM). Both main effects and interactions were examined. For significant ANOVAs, data were further analyzed for multiple comparisons using the least-significant difference (LSD) test. Greenhouse-Geisser epsilon ($\varepsilon$) values were employed when the Greenhouse-Geisser correction was necessary. Estimations of the effect size for ANOVAs were determined with partial $\eta^2$ (partial $\eta^2 = 0.20$ is a small effect size, 0.50 is a medium effect size and 0.80 is a large effect size). Furthermore, Spearman's correlation test was used to calculate the correlation between ApEn or PE values and stage durations. In addition, the Mann–Whitney U-test was used to compare stage durations between sexes for Stages II and III, respectively. SPSS software (release 21) was utilized for the statistical analysis. A significance level of $p < 0.05$ and a high significance level of $p < 0.001$ were used for all comparisons.

## RESULTS

### ApEn and PE values in different anesthesia-awakening stages

For ApEn values, ANOVA analysis (Table 1) revealed significant main effects for the factors "stage" ($F_{2,24} = 25.127$; $p = 0.000$; partial $\eta^2 = 0.677$) and "sex" ($F_{1,12} = 5.033$; $p = 0.045$; partial $\eta^2 = 0.295$) but marginally non-significant effect for the factor "brain area" ($F_{5,60} = 2.373$; $p = 0.050$; partial $\eta^2 = 0.165$), but no significant interaction between factors was observed. ApEn values in Stage III (the recovery stage) were significantly lower than those in both Stages I (the awake stage) and IV (post-anesthesia stage) ($p < 0.05$), however there was no significant difference between Stages I and IV (Table 1 and Fig. 2). In addition, ApEn values for females were significantly higher than the values for males (Table 1).

For PE values, ANOVA analysis (Table 2) showed significant main effects for the factors "stage" ($F_{2,24} = 10.489$; $p = 0.001$; partial $\eta^2 = 0.466$), "sex" ($F_{1,12} = 7.895$; $p = 0.016$; partial $\eta^2 = 0.397$) and "brain area" ($F_{5,60} = 10.584$; $p = 0.000$; partial $\eta^2 = 0.469$). Similar to ApEn, PE values in Stage III (the recovery stage) were significantly lower than those in both in Stages I (the awake stage) and IV (post-anesthesia stage) ($p < 0.05$), however there was no significant difference between Stages I and IV (Table 2 and Fig. 2). Similarly, PE values for females were significantly higher than PE values for males (Table 2). In addition,

**Table 1  Results of ANOVA for ApEn as a function of the factors "stage", "sex" and "brain area".**

| Factors | $F$ | $\varepsilon$ | $p$ | partial $\eta^2$ | LSD |
|---|---|---|---|---|---|
| Stage | $F_{2,24} = 25.127$ | NA | $0.000^{**}$ | 0.677 | I, IV > III |
| Sex | $F_{1,12} = 5.033$ | NA | $0.045^{*}$ | 0.295 | Female > Male |
| Brain area | $F_{5,60} = 2.373$ | NA | 0.050 | 0.165 | NA |
| Stage * Sex | $F_{2,24} = 2.315$ | NA | 0.120 | 0.162 | NA |
| Brain area * Sex | $F_{5,60} = 0.903$ | NA | 0.485 | 0.070 | NA |
| Stage * Brain area | $F_{10,120} = 2.554$ | 0.364 | 0.057 | 0.175 | NA |
| Stage * Sex * Brain area | $F_{10,120} = 1.396$ | NA | 0.190 | 0.104 | NA |

**Notes.**

The symbols '>' denote that approximate entropy (ApEn) values for the given condition on the left side of '>' are significantly larger than those on the right side, and no significant difference exists among the corresponding conditions on the same side of '>' for each case.

Abbreviations: $F$, the $F$ value from ANOVA; $\varepsilon$, the values of epsilon of the Greenhouse-Geisser correction; partial $\eta^2$, effect size for ANOVA; LSD, least-significant difference test; I, Stage I; II, Stage II; III, Stage III; NA, not applicable.

$^{*}p < 0.05$.
$^{**}p < 0.001$.

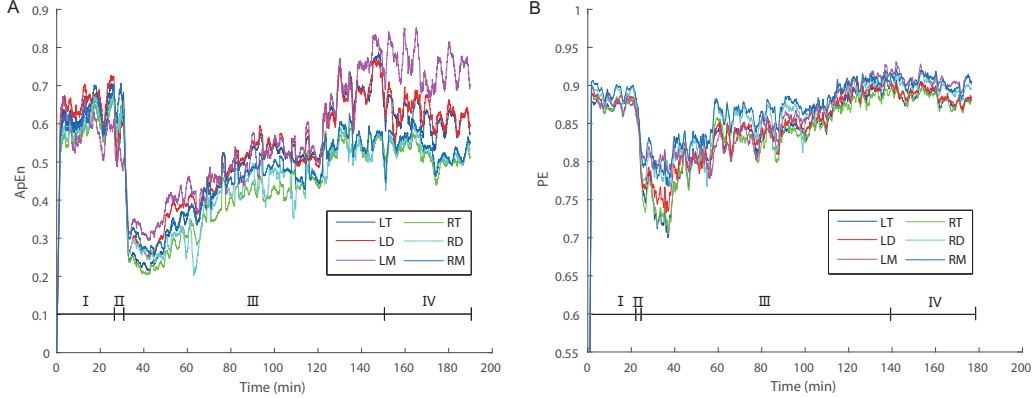

**Figure 2  The dynamic variations of approximate entropy (ApEn) values (A) and permutation entropy (PE) values (B) for the six brain regions for a randomly selected individual.** Note that epochs with artifact were not included. Since the time windows for ApEn (2 s) and PE (10 s) were different, the duration of a given artifact free epoch for the former was shorter than for the latter; thus, the durations of each stage for the former are longer than for the latter. Abbreviations: LT and RT, the left and right telencephalon; LD and RD, the left and right diencephalon; LM and RM, the left and right mesencephalon; I, Stage I (pre-anesthesia stage); II, Stage II (administration stage); III, Stage III (recovery stage); IV, Stage IV (post-anesthesia stage).

PE values for the right mesencephalon were significantly higher than those for other brain areas, while PE values for both sides of the diencephalon and the left mesencephalon were significantly higher than those for the left telencephalon, and that PE values for the right diencephalon were significantly higher than those for the right telencephalon (Table 2).

Unlike ApEn, the interaction between the factors "stage" and "brain area" was significant for PE ($F_{10,120} = 6.665$; $p = 0.000$; partial $\eta^2 = 0.357$; Table 2). Simple effects analysis showed that PE values for the right mesencephalon during a given stage were significantly higher than those for other brain areas ($F_{5,65} = 6.583$, $\varepsilon = 0.480$, $p = 0.003$, partial

**Table 2  Results of ANOVA for PE as a function of the factors "stage", "sex" and "brain area".**

| Factors | $F$ | $\varepsilon$ | $p$ | partial $\eta^2$ | LSD |
|---|---|---|---|---|---|
| Stage | $F_{2,24} = 10.489$ | NA | $0.001^*$ | 0.466 | I, IV > III |
| Sex | $F_{1,12} = 7.895$ | NA | $0.016^*$ | 0.397 | Female > Male |
| Brain area | $F_{5,60} = 10.584$ | NA | $0.000^{**}$ | 0.469 | RM > LT, RT, LD, RD, LM LD, RD, LM > LT RD > RT |
| Stage * Sex | $F_{2,24} = 0.961$ | NA | 0.397 | 0.074 | NA |
| Brain area * Sex | $F_{5,60} = 0.512$ | NA | 0.766 | 0.041 | NA |
| Stage * Brain area | $F_{10,120} = 6.665$ | NA | $0.000^{**}$ | 0.357 | See Table 3 |
| Stage * Sex * Brain area | $F_{10,120} = 1.331$ | NA | 0.222 | 0.100 | NA |

**Notes.**
The symbols '>' denote that permutation entropy (PE) values for the given condition on the left side of '>' are significantly larger than those on the right side, and no significant difference exists among the corresponding conditions on the same side of '>' for each case.

Abbreviations: $F$, the $F$ value from ANOVA; Partial $\eta^2$, effect size for ANOVA; $\varepsilon$, the values of epsilon of the Greenhouse-Geisser correction; LSD, least-significant difference test; I, Stage I; III, Stage III; IV, Stage IV; LT, left telencephalon; RT, right telencephalon; LD, left diencephalon; RD, right diencephalon; LM, left mesencephalon; RM, right mesencephalon; NA, not applicable.

$^*p < 0.05$.
$^{**}p < 0.001$.

$\eta^2 = 0.336$ in Stage I; $F_{5,65} = 8.642$, $p = 0.000$, partial $\eta^2 = 0.399$ in Stage III; $F_{5,65} = 13.395$, $p = 0.000$, partial $\eta^2 = 0.507$ in Stage IV; Table 3) although the difference between the left and right mesencephalon did not reach statistical significance. In addition, PE values for the right diencephalon during Stage III (the recovery stage) were significantly higher than those for both sides of the telencephalon while PE values for the left diencephalon during stages I and IV were significantly higher than values for the left telencephalon. For each brain area, PE values for Stages I and IV were significantly higher than that in Stage III ($F_{2,26} = 13.942$, $p = 0.000$, partial $\eta^2 = 0.517$ for the left telencephalon; $F_{2,26} = 12.234$, $p = 0.000$, partial $\eta^2 = 0.485$ for the right telencephalon; $F_{2,26} = 10.603$, $p = 0.000$, partial $\eta^2 = 0.449$ for the left diencephalon; $F_{2,26} = 5.926$, $p = 0.008$, partial $\eta^2 = 0.313$ for the right diencephalon; $F_{2,26} = 9.289$, $p = 0.001$, partial $\eta^2 = 0.417$ for the left mesencephalon; $F_{2,26} = 6.507$, $p = 0.005$, partial $\eta^2 = 0.334$ for the right mesencephalon; Table 3), although the difference between Stages III and IV did not reach statistical significance for the right diencephalon. Moreover, PE values for Stage I was higher than that in Stage III for both sides of the telencephalon and the left diencephalon.

## Stage durations vs. ApEn or PE values

Correlation analysis was used to determine whether the durations of the anesthesia stages (Stages II and III) were associated with ApEn or PE values for each brain area and each stage. For ApEn measurement, significant correlations between stage durations and ApEn values were found exclusively for the right hemisphere, especially for the right thalamus (Table 4 and Fig. 3). ApEn values in Stage I were positively correlated with the duration of Stage II for the right diencephalon, i.e., the right thalamus ($r = 0.534$, $p = 0.049$;

**Table 3  Results of simple effects analysis for PE as a function of the factors "stage" and "brain area".**

| Factors | $F$ | $\varepsilon$ | $p$ | partial $\eta^2$ | LSD |
|---|---|---|---|---|---|
| *Stage* | | | | | |
| I | $F_{5,65} = 6.583$ | 0.480 | 0.003[*] | 0.336 | RM>LT, RT, LD, RD, LM LD >LT |
| III | $F_{5,65} = 8.642$ | NA | 0.000[**] | 0.399 | RM>LT, RT, LD, RD, LM RD >LT, RT |
| IV | $F_{5,65} = 13.395$ | NA | 0.000[**] | 0.507 | RM>LT, RT, LD, RD LM>LT, RT, LD LD>LT |
| *Brain area* | | | | | |
| LT | $F_{2,26} = 13.942$ | NA | 0.000[**] | 0.517 | I>IV>III |
| RT | $F_{2,26} = 12.234$ | NA | 0.000[**] | 0.485 | I>IV>III |
| LD | $F_{2,26} = 10.603$ | NA | 0.000[**] | 0.449 | I>IV>III |
| RD | $F_{2,26} = 5.926$ | NA | 0.008[*] | 0.313 | I>III |
| LM | $F_{2,26} = 9.289$ | NA | 0.001[*] | 0.417 | I, IV>III |
| RM | $F_{2,26} = 6.507$ | NA | 0.005[*] | 0.334 | I, IV > III |

Notes.

The symbols '>' denote that permutation entropy (PE) values for the given condition on the left side of '>' are significantly larger than those on the right side, and no significant difference exists among the corresponding conditions on the same side of '>' for each case.

Abbreviations: $F$, the F value from ANOVA; Partial $\eta^2$, effect size for ANOVA; $\varepsilon$, the values of epsilon of the Greenhouse-Geisser correction; LSD, least-significant difference test; I, Stage I; III, Stage III; IV, Stage IV; LT, left telencephalon; RT, right telencephalon; LD, left diencephalon; RD, right diencephalon; LM, left mesencephalon; RM, right mesencephalon; NA, not applicable.

[*]$p < 0.05$.

[**]$p < 0.001$.

Table 4 and Figs. 3A–3C), while ApEn values in Stage III were negatively correlated with the duration of Stage III for the right hemisphere ($r = -0.591$, $p = 0.026$ for the right telencephalon; $r = -0.600$, $p = 0.023$ for the right thalamus; $r = -0.552$, $p = 0.041$ for the right mesencephalon; Table 4 and Figs. 3D–3F). Moreover, ApEn values in Stage IV were negatively correlated with the duration of Stage III for the right thalamus ($r = -0.609$, $p = 0.021$; Table 4 and Fig. 3H). Similar to ApEn, significant correlations between stage durations and PE values were found exclusively for right brain areas (Table 5 and Fig. 4). PE values in Stage II were negatively correlated with the duration of Stage III for the right thalamus and the right mesencephalon ($r = -0.670$, $p = 0.009$ for the former; $r = -0.543$, $p = 0.045$ for the latter; Table 5 and Figs. 4B–4C), while PE values in Stage III were negatively correlated with the duration of Stage III for the right hemisphere ($r = -0.578$, $p = 0.030$ for the right telencephalon; $r = -0.543$, $p = 0.045$ for the right thalamus; $r = -0.675$, $p = 0.008$ for the right mesencephalon; Table 5 and Figs. 4D–4F). In addition, the duration of the administration stage for females was significantly longer than that for males ($U = 7$, $p = 0.025$; Table 6), while the duration of the recovery stage for female was shorter than that for males ($U = 13$, $p = 0.142$; Table 6).
**Table 4  Results of the correlation analysis between ApEn and the duration of a given stage.**

| | LT | | RT | | LD | | RD | | LM | | RM | |
|---|---|---|---|---|---|---|---|---|---|---|---|---|
| | r | p | r | p | r | p | r | p | r | p | r | p |
| Duration (II) vs. ApEn (I) | 0.446 | 0.110 | 0.525 | 0.054 | 0.424 | 0.131 | 0.534 | 0.049* | 0.433 | 0.122 | 0.481 | 0.081 |
| Duration (II) vs. ApEn (II) | −0.068 | 0.817 | 0.196 | 0.503 | −0.002 | 0.994 | 0.305 | 0.288 | 0.231 | 0.427 | 0.284 | 0.326 |
| Duration (II) vs. ApEn (III) | 0.253 | 0.383 | −0.051 | 0.864 | 0.182 | 0.533 | −0.042 | 0.887 | 0.323 | 0.260 | −0.042 | 0.887 |
| Duration (II) vs. ApEn (IV) | −0.042 | 0.887 | −0.046 | 0.876 | −0.222 | 0.446 | −0.305 | 0.288 | 0.112 | 0.703 | 0.029 | 0.923 |
| Duration (III) vs. ApEn (I) | −0.077 | 0.794 | −0.165 | 0.573 | −0.112 | 0.703 | −0.231 | 0.427 | 0.002 | 0.994 | −0.191 | 0.513 |
| Duration (III) vs. ApEn (II) | −0.248 | 0.392 | −0.204 | 0.483 | −0.499 | 0.069 | −0.266 | 0.358 | −0.433 | 0.122 | −0.310 | 0.281 |
| Duration (III) vs. ApEn (III) | −0.380 | 0.180 | −0.591 | 0.026* | −0.407 | 0.149 | −0.600 | 0.023* | −0.442 | 0.114 | −0.552 | 0.041* |
| Duration (III) vs. ApEn (IV) | −0.090 | 0.759 | −0.495 | 0.072 | −0.051 | 0.864 | −0.609 | 0.021* | −0.051 | 0.864 | −0.323 | 0.260 |

**Notes.**

Abbreviations: $r$, correlation coefficient; LT, left telencephalon; RT, right telencephalon; LD, left diencephalon; RD, right diencephalon; LM, left mesencephalon; RM, right mesencephalon; I, Stage I; II, Stage II; III, Stage III; IV, Stage IV.

*$p < 0.05$.

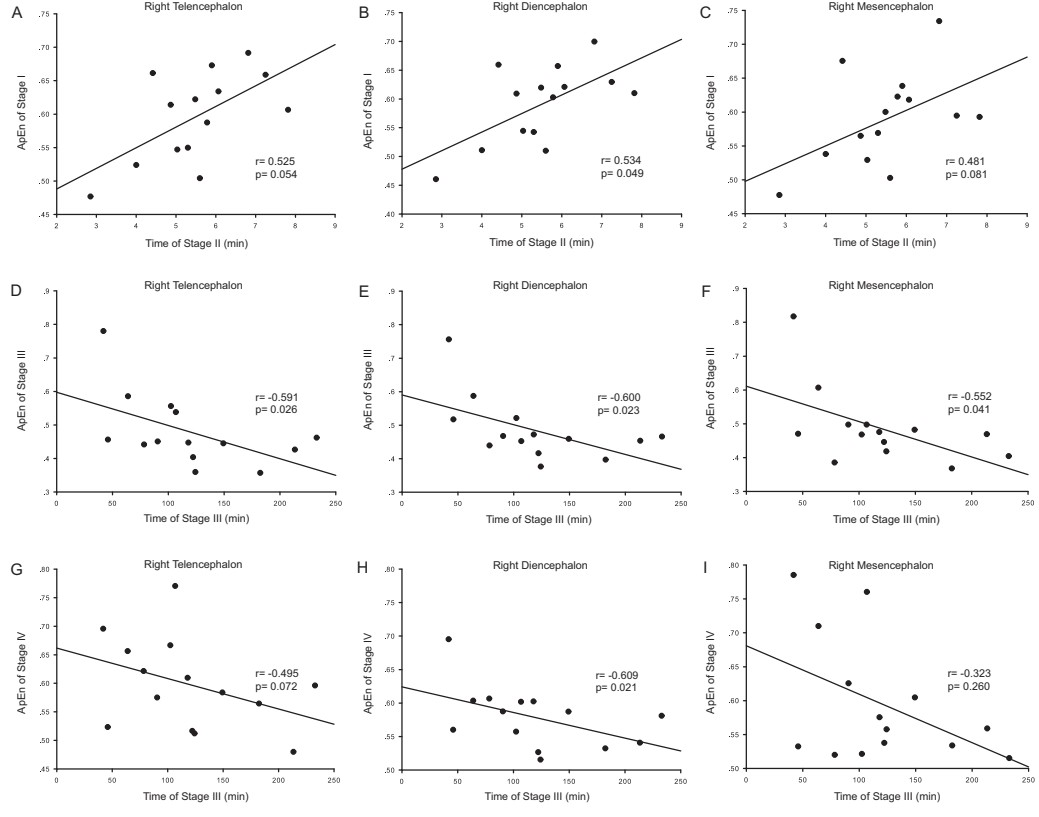

**Figure 3  Correlation analysis between ApEn and the duration of a given stage for the right hemisphere.** The correlation between the duration of Stage II and the approximate entropy (ApEn) values of brain structures during Stage I for the right hemisphere (A–C), the correlation between duration of Stage III and the ApEn values of Stage III for the right hemisphere (D–F), and the correlation between the duration of Stage III and the ApEn values of Stage IV for the right hemisphere (G–I).

**Table 5  Results of the correlation analysis between PE and the duration of a given stage.**

| | LT | | RT | | LD | | RD | | LM | | RM | |
|---|---|---|---|---|---|---|---|---|---|---|---|---|
| | r | p | r | p | r | p | r | p | r | p | r | p |
| Duration (II) vs. PE (I) | 0.451 | 0.106 | 0.354 | 0.215 | 0.358 | 0.208 | 0.130 | 0.659 | 0.130 | 0.659 | −0.160 | 0.584 |
| Duration (II) vs. PE (II) | 0.248 | 0.392 | 0.477 | 0.085 | 0.130 | 0.659 | 0.446 | 0.110 | 0.371 | 0.191 | 0.248 | 0.392 |
| Duration (II) vs. PE (III) | 0.349 | 0.221 | −0.015 | 0.958 | 0.024 | 0.935 | −0.275 | 0.342 | 0.182 | 0.533 | −0.147 | 0.615 |
| Duration (II) vs. PE (IV) | 0.081 | 0.782 | 0.138 | 0.637 | 0.200 | 0.493 | −0.011 | 0.970 | 0.130 | 0.659 | −0.191 | 0.513 |
| Duration (III) vs. PE (I) | 0.046 | 0.876 | 0.086 | 0.771 | 0.218 | 0.455 | 0.046 | 0.876 | 0.134 | 0.648 | −0.007 | 0.982 |
| Duration (III) vs. PE (II) | −0.187 | 0.523 | −0.530 | 0.051 | −0.380 | 0.180 | −0.670 | 0.009* | −0.455 | 0.102 | −0.543 | 0.045* |
| Duration (III) vs. PE (III) | −0.521 | 0.056 | −0.578 | 0.030* | −0.490 | 0.075 | −0.543 | 0.045* | −0.495 | 0.072 | −0.675 | 0.008* |
| Duration (III) vs. PE (IV) | −0.231 | 0.427 | −0.468 | 0.091 | −0.130 | 0.659 | −0.407 | 0.149 | −0.143 | 0.626 | −0.477 | 0.085 |

**Notes.**

Abbreviations: $r$, correlation coefficient; LT, left telencephalon; RT, right telencephalon; LD, left diencephalon; RD, right diencephalon; LM, left mesencephalon; RM, right mesencephalon; I, Stage I; II, Stage II; III, Stage III; IV, Stage IV.

*$p < 0.05$.

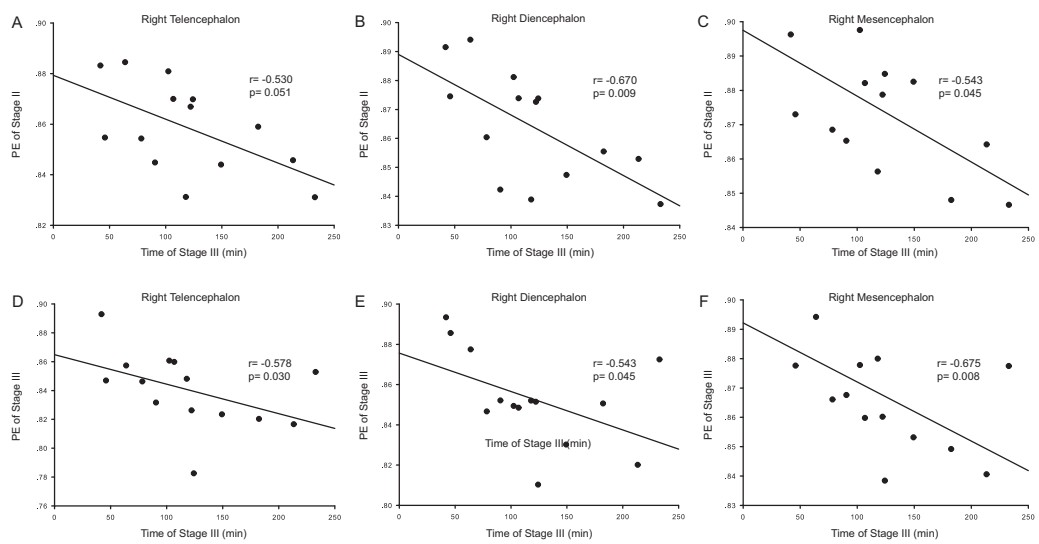

**Figure 4  Correlation analysis between PE and the duration of a given stage for the right hemisphere.** The correlation between the duration of Stage III and the Permutation entropy (PE) values of brain structures during Stage II for the right hemisphere (A–C), and the correlation between duration of Stage III and the PE values of Stage III for the right hemisphere (D–F).

## DISCUSSION

### Anesthesia-awakening cycle in ApEn or PE variation

During the awake period (Stage I) animals remain vigilant. Consistent with this the highest ApEn or PE values reflect the demands placed on the brain for processing internal and external stimuli accurately. During the administration stage (Stage II), the animals gradually became motionless and ApEn or PE values decreased sharply (Fig. 2). At the point that ApEn or PE decreased to the lowest level the animals became apparently unconscious insofar as they exhibited no response to the limb pinching stimulus. During the recovery

**Table 6** Means and standard deviations of the durations of Stages II and III for females and males.

| Stages | Female (min) | Male (min) | $p$ |
|---|---|---|---|
| II | $6.25 \pm 1.14$ | $4.78 \pm 1.03$ | 0.025[*] |
| III | $99.47 \pm 59.14$ | $139.33 \pm 53.95$ | 0.142 |

Notes.

This table does not include Stages I and IV because the durations of these two stages were constant, i.e., 30 min.

Abbreviations: II, the administration stage; III, the recovery stage.

[*]$p < 0.05$.

stage (Stage III) ApEn and PE values increased gradually and the toe pinch response reappeared.

ApEn and PE typically reflects the complexity and regularity of brain activity (*Burioka et al., 2005*; *Liang et al., 2015*). High ApEn or PE values indicate high complexity, random and unpredictable changes, whereas low ApEn or PE values indicate low complexity, regularity and predictability of EEG signals (*Burioka et al., 2005*). In this study, we analyzed changes in the ApEn and PE of EEG signals during the "pre-anesthesia, administration, recovery, and post-anesthesia stages" cycle in *X. laevis* frogs, and found that there were significant differences in ApEn or PE among these stages. ApEn and PE values were the highest when awake (pre-anesthesia and post-anesthesia stages) and lowest during the administration stage (Tables 1 and 2; Fig. 2).

The occurrence of the highest ApEn or PE values during the stages the frogs were awake is consistent with the idea that at this time complex tasks (such as accurate perception and the processing of internal and external stimuli) can be carried out by the brain (*Heinke & Koelsch, 2005*). EEG ApEn and PE show a sharp decrease around the point of loss-of-responsiveness during the administration stage, consistent with previous studies in humans reporting that the transition into anesthetic unconsciousness is associated with dramatic and abrupt changes in the population-average membrane voltage in cortical neurons (*Heinke & Koelsch, 2005*). The present results indicate that both ApEn and PE values reflect changes in the complexity of amphibian brain activities similar to those found in human patients (*Bruhn, Röpcke & Hoeft, 2000*) across different stages in the anesthesia-awakening cycle.

## Right thalamus may play an important role in anesthesia-awakening regulation

Both ApEn and PE can track EEG changes associated with different anesthesia states although PE performs best; ApEn performs best in detecting neuron burst suppression (*Liang et al., 2015*). Consistent with this, the main effects of the present results were similar for ApEn and PE. In the present study, different anesthesia states manifested as behavioral changes that would be expected to be closely related to neuron burst suppression, a phenomenon that would physiologically reflect possible functional lateralization for general anesthesia regulation. For this reason, the following discussion is mainly based on the ApEn results.

During Stage I (the pre-anesthesia stage), ApEn values in the right thalamus were positively correlated with the duration of Stage II (the administration stage). This means that as the complexity of right thalamic neural network activity increases during the pre-anesthesia stage, induction time for general anesthesia increases. Since the complexity of the EEG represents the activity level of specific brain regions, this correlation shows that the right thalamus exhibits significant EEG pattern changes during the anesthesia-awakening cycle. The results also show an inverse correlation between right thalamic ApEn values in Stage IV and the duration of Stage III. This means that increased inhibition of right hemispheric neural information processing activity results in longer recovery times and that correspondingly, longer recovery times are associated with more inhibited thalamic activity in the post-anesthesia stage. In addition, there was an inverse correlation between right hemispheric ApEn values in Stage III and the duration of this stage, suggesting that maintenance of higher activities in the right hemisphere (including the right thalamus) during the recovery stage increases the likelihood that the subject will easily awaken. It is therefore notable that the significant correlations between EEG ApEn and PE values during the anesthesia-awakening cycle appeared primarily for the right diencephalon in the present study (Tables 4 and 5; Figs. 3 and 4). Thus, it seems reasonable to hypothesize that the right thalamus plays a key regulatory role in the anesthesia-awakening cycle.

The above described right-lateralized effects could not have resulted from the fact that the habenular nuclei in amphibians are markedly asymmetrical in size insofar as the left habenular complex consists of two distinct nuclei whereas the right habenular nucleus consists of only one cell group (Schmidt, 1976). The habenular nucleus is a small bilateral structure, located in the anterior dorsal diencephalon (i.e., the epithalamus), behind the pineal gland, on either side of the third ventricle (Kemali & Braitenberg, 1969). In the present study, the two diencephalic electrodes were situated immediately above the dorsal thalamus which is the main source of subcortical inputs to the cerebral cortex (Harris, Guglielmotti & Bentivoglio, 1996). In other words, asymmetric habenular nuclei could not be responsible for the right-lateralized effects observed in the present results, consistent with the idea that the right thalamus is of key importance in anesthesia-awakening regulation in frogs.

The vertebrate CNS consists of anatomically and functionally distinct regions which also differ in sensitivity to anesthetics (Heinke & Koelsch, 2005). Several sites in the brain including the cerebral cortex, thalamus, limbic system and reticular formation have been proposed to play key roles in the regulation of consciousness with the thalamo-cortical circuitry as critically important for controlling consciousness and attention (Heinke & Koelsch, 2005; Heinke & Schwarzbauer, 2002). The thalamus serves as a relay or a gate for much of the sensory information projected to the telencephalon including the cortico-thalamo-cortical pathway (Béhuret et al., 2013; Sherman & Guillery, 2002). Thus, as a center of the brain, the thalamus has been characterized as a compact 'miniature map' of the rest of the brain (Ward, 2011). Consistent with this idea, thalamic lesions can result in profound cognitive disorders including delirium, aphasias, confusion, unconsciousness and even death (Llinas et al., 1998; Ward, 2013). Recent studies using PET and fMRI support the idea that the thalamus functions as a macroscopic locus (target) in which

general anesthetics bring about the unconscious state in patients (*Heinke & Koelsch, 2005*) and which is normally involved in regulating cortical arousal and the activities of cortical networks (*McCormick & Bal, 1997*; *Schiff, 2008*; *Steriade, 1996*). Moreover, anesthetic-induced unconsciousness is consistently associated with a reduction in metabolism or blood flow in the thalamus, which indicates that the thalamus can act as a consciousness switch (*Alkire et al., 1997*; *Fiset et al., 1999*). Our results suggest that the thalamus is the crucial region for anesthesia-induced unconsciousness in frogs and regulates the amphibian anesthesia-awakening cycle, similar to the condition in humans (*Xie et al., 2011*).

Although the role of the thalamus in conveying sensory input to the telencephalon is firmly established in reptiles (*Pritz, 2016*), the cortico-thalamo-cortical loop remains to be identified. However, human studies have shown that the loop modulates unconsciousness under general anesthetics (*Fiset et al., 1999*) because arousal is elicited by the brainstem-thalamus-cortex activating system (*Steriade, 1996*). As a central node of these brain networks, the thalamus plays an important role in supporting consciousness in two ways. First, specific thalamic nuclei relay sensory and motor messages that may become part of the content of consciousness. Second, nonspecific thalamic nuclei are likely involved in the control of cortical arousal originating from the brainstem reticular formation (*Zhou et al., 2011*).

The present results show that the right thalamus is physiologically positioned to play a regulatory role in anesthesia-awakening cycle. This is consistent with the idea that both the structural and functional lateralization of nervous system function is conserved throughout vertebrates including humans (*Rogers & Vallortigara, 2015*; *Samara & Tsangaris, 2011*; *Vallortigara & Versace, 2017*) and is manifest in many invertebrates. Examples include preferential use of the right or left hemispheres during various information processing tasks and behaviors in humans, primates, birds, reptiles, amphibians, fish, bees, fruit flies and nematodes (*Fang et al., 2015*; *Frasnelli, Vallortigara & Rogers, 2012*; *Lippolis et al., 2002*; *Robins & Rogers, 2006*; *Rogers, 2014*; *Rogers et al., 2013*; *Rosa-Salva et al., 2012*; *Roussigné, Blader & Wilson, 2012*; *Vallortigara et al., 1998*; *Vallortigara & Versace, 2017*; *Xue et al., 2015*). For example, the right hemisphere is usually dominant for spatial attention in humans, while the left preferentially processes language and formal reasoning (*Geschwind & Miller, 2001*). Brain asymmetry is thought to be a conserved and fundamental feature which enhances the efficiency of information processing, so that functional specialization of one hemisphere frees the contralateral hemisphere to perform other tasks (*Rogers & Vallortigara, 2015*; *Vallortigara & Rogers, 2005*).

Brain anatomical asymmetries are the basis of functional lateralization (*Samara & Tsangaris, 2011*). The right thalamus is larger than the left in humans (*Péran et al., 2009*; *Sullivan et al., 2004*). Moreover, lesions of the left and right thalamus typically do not result in the same language deficits (*Ojemann, 1977*). Thus, it seems reasonable to speculate that the right thalamus would be less likely to be suppressed by general anesthetics. This speculation is consistent with our current results showing that both ApEn and PE values in the right thalamus were higher than the left counterpart during the awakening period and anesthesia period, although these differences did not reach statistical significance. In other words, the dynamic neural activity changes in the right thalamus would be expected

to be smaller than those in the left during anesthesia, consistent with its possible regulation function in the anesthesia-awakening cycle. However, it remains for future research to explicate the causal basis of the right thalamic regulation function including the anatomical and neurophysiological characteristics underlying its apparently important role.

## Sex differences in general anesthesia

The current results show that both ApEn and PE values for females are higher than those for males, the duration of the administration stage for females is significantly longer than males, while the duration of the recovery stage for females is shorter than males. These results indicate that female frogs are less sensitive to the hypnotic effect of anesthetics than males, consistent with clinical studies in humans which have shown that women exhibit higher bispectral index values (*Buchanan, Myles & Cicuttini, 2011*) than men under the same anesthetic dosage and usually show faster recovery after the administration of anesthetics compared with men (*Buchanan, Myles & Cicuttini, 2011*; *Buchanan et al., 2006*).

Sexually dimorphic behaviors in mammals typically reflect sex differences in the nature of reciprocal influences among genes, gonadal functions, hormones and environmental factors (*Kelly, Ostrowski & Wilson, 1999*). Some sex hormones are neurosteroids with anesthetic properties. For instance estradiol and progesterone are involved in general anesthesia regulation and post-anesthesia recovery (*Buchanan et al., 2006*). High doses of progesterone can anesthetize animals and humans (*Bitran, Purdy & Kellog, 1993*; *Merryman et al., 1954*). Previous research suggests that progesterone and estradiol affect anesthesia maintenance and recovery by influencing the excitability of neurons in the brain and brainstem (*Woolley & Schwartzkroin, 1998*). Thus, the general anesthesia state depends on the interactions of anesthetics with different receptors in the brain, and sex differences in general anesthesia and post-anesthesia recovery in many animal species may be related to the levels of sex hormones and their distribution (*Buchanan, Myles & Cicuttini, 2011*). It thus seems reasonable to speculate that these phenomena include amphibians as well.

On the other hand, previous studies in fishes and amphibians have found that larger subjects need longer MS-222 induction times and exhibit shorter recovery times (*Cecala & Price, 2007*; *Paduano et al., 2013*; *Zahl et al., 2009*; *Zahl et al., 2011*). In the present study, the body mass of females on average was twice that of males, suggesting that the observed sex differences may have resulted from body mass differences. Future research is required to determine whether the combined actions of sex hormones and body mass differences bring about the sex differences in general anesthesia observed here.

In summary, both ApEn and PE measurements are suitable for estimating the complexity of amphibian brain activity, the right thalamus appears well positioned physiologically to play an important role in anesthesia-awakening regulation in frogs, and sex differences exist in the neural regulation of general anesthesia in frogs.

## ACKNOWLEDGEMENTS

The authors gratefully acknowledge all the members of the Behavioral Neuroscience Group for their discussion and help for the experiments reported here.

### Funding
This work was supported by a grant from the National Natural Science Foundation of China (No. 31672305 and No. 31372217) to Guangzhan Fang. The funders had no role in study design, data collection and analysis, decision to publish, or preparation of the manuscript.

### Grant Disclosures
The following grant information was disclosed by the authors:
National Natural Science Foundation of China: 31672305, 31372217.

### Competing Interests
The authors declare there are no competing interests.

### Author Contributions
- Yanzhu Fan conceived and designed the experiments, performed the experiments, analyzed the data, contributed reagents/materials/analysis tools, prepared figures and/or tables, authored or reviewed drafts of the paper, approved the final draft.
- Xizi Yue and Fei Xue performed the experiments, analyzed the data, contributed reagents/materials/analysis tools, authored or reviewed drafts of the paper, approved the final draft.
- Steven E. Brauth and Yezhong Tang conceived and designed the experiments, authored or reviewed drafts of the paper, approved the final draft.
- Guangzhan Fang conceived and designed the experiments, analyzed the data, prepared figures and/or tables, authored or reviewed drafts of the paper, approved the final draft.

### Animal Ethics
The following information was supplied relating to ethical approvals (i.e., approving body and any reference numbers).

The Animal Care and Use Committee of Chengdu Institute of Biology, Chinese Academy of Sciences provided approval for this study.

### Supplemental Information
Supplemental information for this article can be found online at http://dx.doi.org/10.7717/peerj.4516#supplemental-information.

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
