# Peer review of "The right thalamus may play an important role in anesthesia-awakening regulation in frogs"

_PeerJ, doi:10.7717/peerj.4516_

## Round 0.1 · original submission · Major Revisions

As you can see, although both Reviewers think that your contribution can result in a publishable paper, they also recommend major revisions as necessary. Please be careful on methods description and in reporting statistics. Also, describe your results cautiously when data suggest only marginally significant effects.

On my side at editorial level, I would recommend some addition/updating on references on brain asymmetry. In particular, Frasanelli (2013) is the shorter version of a major review published elsewhere (Frasnelli et al. (2012). Left-right asymmetries of behaviour and nervous system in invertebrates. Neuroscience and Biobehavioral Reviews, 36: 1273-1291) I would recommend quoting the latter.
Also, some more updated reviews are needed (I would recommend the following but others are available in the literature: Rogers et al. (2013) Divided Brains. Cambrdige Univ. Press;
Vallortigara, G., Versace, E. (2017). Laterality at the Neural,
Cognitive, and Behavioral Levels. In “APA Handbook of Comparative Psychology: Vol. 1. Basic Concepts, Methods, Neural Substrate, and Behavior”, J. Call (Editor-in-Chief), pp. 557-577, American Psychological Association, Washington DC.

Reviewer 1 ·

Basic reporting

The Right Thalamus may Play an Important Role in Anesthesia-Awakening Regulation in Frogs

The present manuscript reports a well-designed simple and clear experiment. Authors aimed to investigate the neural correlates underlying the state of General Anesthesia in an amphibian species and to test whether its regulation is lateralized in the thalamus. Authors devised a single experiment in which the complexity of brain activity (estimated using approximate entropy) was recorded in fourteen frogs (Xenopus laevis), implanted with six EEG recording electrodes (three per each hemisphere positioned in: telencephalon, diencephalon and mesencephalon) during four different stages (before, during and after administration of anesthesia). Authors report main effects of stage and sex for the level of approximate entropy. Indeed higher level of complexity were observed before and after recovery from anesthesia and those levels were significantly higher than during the recovery stage. Moreover, they showed that overall females exhibited significantly higher levels of estimated approximate entropy than males. The level of complexity measured before anesthesia positively correlated with the time anesthesia required being effective only in the right hemisphere. The level of complexity measured in the right hemisphere during the administration stage negatively correlated with the recovery time. Eventually, a negative correlation was found between the complexity measured in the right diencephalon only and the recovery time. From these results, authors conclude that thalamus (part of diencephalon) seems to have a distinctive asymmetrical role in anesthesia-awakening regulation in frogs, suggesting an old phylogenetic origin in the regulation of arousal and consciousness.
MAJOR COMMENTS:
1. In the Introduction, authors state that non-linear behaviors exhibited from neurons can only be adequately described using non-linear parameters (starting from line 74), “Among these methods” (line 76-77) they chose approximate entropy to estimate complexity of the EEG signals. It would be extremely useful to provide the reader with more information about other nonlinear methods used for estimating the complexity of EEG signal and particularly why among all the others approximate entropy was chosen. What are the characteristics of this method that made authors chose it instead of choosing others? What are the advantages of using approximate entropy in this work instead of using other nonlinear parameters?
2. In line with previous point, in section Materials and Methods, authors should stress the advantages of using approximate entropy instead of other nonlinear methods. I would strongly recommend providing a second nonlinear statistical method and applying it to raw data in parallel to approximate entropy. I believe this could increase enormously the validity of the findings. Comparisons between results obtained with the two different methods would enhance authors’ conclusions.
3. Results section is difficult to follow, too concise and lacks in coherence with the statistical analyses’ part. Results should be reported in order of relevance from the main effects to the minor ones. Results should be compared between significant and non-significant one. Moreover, the text in the results section should only describe what was observed and not explain it that pertains to the Discussion section. I highly recommend re-writing it completely.
4. Lines 284-292 in the Discussion section look odd and it is difficult to follow, the reason why habenular nuclei were discussed is put at the end of the paragraph and the reader risks to lose the point. I strongly suggest rephrasing it.
5. In the Discussion section within the “Sex differences in GA” paragraph (starting from line 348), authors discuss the differences found between sex from a comparative perspective with humans. Indeed, sexual differences exist in the effectiveness of anesthesia between males and females in humans and the same differences are found in frogs. Authors argue that sex hormones may play a role in the sex differences found in frogs as well as they play it in humans. “The results of the present study indicate these phenomena include amphibians as well.” (lines 367-368). I would highly recommend the authors to stress that this a speculative conclusion. Moreover, the raw data provide a more parsimonious explanation of these sex differences in the susceptibility to anesthesia (higher doses needed for female and higher recovery time for males), by looking at the weight of subjects it is clear that females on average weight twice a male. This information suggest that difference in the dosage and in the recovery time may be dependent on subjects’ weight rather than on their gender.
6. I would avoid adding a section Conclusion if it is so short, I would suggest instead adding it as a summary part of the discussion.

MINOR COMMENTS:

a. Lines 43-44 “Previous studies have indicated that specific brain regions are more sensitive to anesthesia..” please provide some examples of the brain regions you refer to.
b. i.e. line 52 use of the verb “poise” please re-consider the word choice.
c. i.e. line 58 paper cited “Salva et al., 2012” the citation is wrong, correct citation is “Rosa-Salva et al., 2012” please change it throughout the text.
d. Line 62 I think a type error occurred and a comma was placed instead of a full stop.
e. Line 84 I believe a reference is missing to support the homologies between the general pattern of connectivity between amniotes and amphibian diencephalon.
f. Line 88 after the full stop I believe a new paragraph should start addressing the choice of anesthesia.
g. Section Material and Methods “Statistical analyses” lines 220-222 the selected levels of significance are stated “A significance level of p < 0.05 was used for all comparisons; p values > 0.05 and < 0.1 were considered as marginally significant (Altman et al., 1983; Utts & Heckard, 2005)”. Whereas in the Results section and in the tables different levels of significance are used, I strongly recommend being consistent throughout text.
h. In the Results section at the beginning, I would suggest authors to point out that from line 225 to line 231 is a descriptive level of analysis.
i. In the section Results values of significant post-hoc tests will be easier to follow if inserted in the main text body where they are presented and not only in the tables, which delays reader understanding.
j. Figure 3 is not easy to read and I believe is not providing new information to the reader. I strongly suggest removing it, or to change it completely.
k. In Figure 3 for instance, different recording sites are labelled as LT, RT, LD…using the letters of the corresponding brain subdivision. Whereas, in the Materials and Methods the same sites are labeled R1, R2, R3.. To enhance the readability I suggest keeping abbreviations consistent throughout the text.
l. I strongly recommend in Figure 4 to add to the picture and not only to the caption to which brain subdivision the plots refer (i.e. Right Telencephalon). Moreover, I suggest to put axis labels for each plot and to write to which stage it refers. I.e., first plot on the upper left side: title of x-axis could be “time stage II” and y-axis “ApEn Stage I”. Figure caption should be complete. Eventually, authors should really re-consider whether the picture is really informative to the reader and helps the understanding or not.
m. In the Discussion section lines 311-313 the reference is referring to humans whereas the sentence is speaking about frogs and amphibians, I believe “as it was described for humans” is missing.
n. Line 360 type error “for instence”.

Experimental design

no comment look above

Validity of the findings

no comment look above

Additional comments

no comment look above

Reviewer 2 ·

Basic reporting

no comment

Experimental design

is very simple

Validity of the findings

Brain region specific findings are not convenient, the differences between stages and a potential role of sex might be fine, but the analysis needs to be worked out and summarised better.

Additional comments

Review of the manuscript

“The right thalamus may play an important role in anesthesia-awakening regulation in frogs (#22293)”

submitted for a publication in PeerJ.

The presented study explores the brain activities in a common laboratory frog species (Xenopus laevis) during different stages of anaesthesia and awakening measured based on EEG recordings from implanted electrodes in different brain areas. The EEG data was analyzed by calculating approximate entropy (ApEn) values, which reflect the complexity and regularity of brain activity. The main finding of the study is, that brain activity regularity and complexity is reduced during a general anaesthesia and is increasing during awaking stages. Moreover, some trends were indicated which reflect brain region specific and sex dependent variations. E.g. some correlations are reported to be significant in the right thalamus between the duration and ApEn for some stages as well as brain activities in females are higher than in males. The importance of thalamus for regulation of consciousness are discussed by considering also the functional lateralisation of the brain and the effect of sex differences.

After a brief reading, as such these results, would be suitable for a publication in PeerJ (given the philosophy of the Journal as far as I remember). However, in my opinion the authors should significantly revise and overwork the manuscript. Although the experiment was very simple, the manuscript in its present form is very difficult to follow, in particular the results section is not very clear, contains unnecessary complexity and at the same time is missing important information’s, which overall makes it very difficult to understand the findings. The authors should also carefully revise the introduction/discussion and check the validity of their statements and corresponding references (E.g. line 315 the authors cite a study on caimans (Pritz 2016) by considering these crocodile species as amphibians). Also, the authors should be more careful with over claims by making interpretations of the overall very weak results. As far as I can see apart from the activity reduction of the brain activity during anaesthesia, the potential effect of factors such as region specificity, sex and lateralisation is very week in the given data. The manuscript should be more focused e.g if the study is intended to validate the ApEn approach, the manuscript should be written as a methodological paper. If it is about lateralisation or the specific functions of the thalamus during anaesthesia, more data need to be collected. However, it is of course possible to report the findings as a it is and to speculate on the potential region-specific functions in the discussion, but the results need to be presented very clear, the main findings should be visible and it must be made clear when it is a speculation about additional potential findings.

Very Important: 5% error probability is a widely-accepted convention and already a very “soft” measurement of significance. I would strongly recommend the authors not to consider any values with p>0.05 as significant! However, it would be acceptable if the values closely p>0.05<0.1 would be discussed as marginally non-significant trends which need to be confirmed in future studies but these trends should not be the core of the main results for a scientific publication and should not be put in the same categories as other significant results.

Other more specific comments are here below:

- Line 39: To many shortenings reduce readability. I would recommend not to shorten ‘general anaesthesia’ into ‘GA’ and instead to write the whole wording in the whole manuscript.
- Line 71: EEG is not the most direct approach, please rephrase e.g. instead of ‘is the most direct approach’ write ‘allows’
- Line 117: instead of ‘…water bath using MS-222 solution (3.5g/L, 500ml)…’ would be better ‘‘…water bath using 500ml of MS-222 solution (3.5g/L)…’
- Line 133, specify how deep the electrodes were implanted
- Line 221, see comment above on the significance of results.
- Line 232, Table 1 should be a figure and contain the results for all four stages (negative results should be reported)
- Line 234-239 and Table 2: Interaction of brain area*stage*sex need to be reported. if there was no such significant interaction, it is at least statistically not allowed to split the data into stages, areas and sex for any post hoc analysis. In such a case the authors should try a simpler approach by removing one of the factors e.g. stage*sex might be potentially promising.
- Line 240-253: these are a lot of correlations, of which as far as I understand only a part of all possible comparisons is reported and even here with only few significant values. If it is true, that the correlations (negative or positive) are more often occurring in the right hemisphere, than the authors should think about how to summarise these results into a simpler and more convenient analysis. In any case the authors should be very careful about the interpretation and discussion of these correlations by making conclusion about the role of specific brain regions.
- Line 244: ‘ApEn values in Stage I were positively correlated with the duration of Stage II ‘. The authors should explain better the potential meaning of such correlations of activities between different stages in the discussion.
- Line 274: As it is stated here ‘…ApEn values in the right thalamus and telencephalon were positively correlated with the duration of Stage II…’, however the discussion is focusing on the specific role of the thalamus, which in my opinion is not so clear based on the data. The authors should be more careful with over claims and identify this part of the discussion as rather speculative.
- Line 291-293: ‘…habenular nuclei could not be responsible….’ is a very strong claim, the explanation in the paragraph line 294-293 does not provide enough evidences for this conclusion. The authors should be more careful with such short cuts in the discussion.
- Line 315: caimans are not amphibians. See also the comment above.
- Figure 3: The y axis should contain the sex e.g. in A: ApEn (males) and B: ApEn (females). Also, please remove any ‘a-d’ form the graphs, here it is very difficult to understand what was compared with what. Instead please make a line above the compared bars with one or two asterisks for significant results. Do not specify non-significant trends here. Overall, it seems that all results among different brain regions are very similar and I am wondering if it is meaningful to divide into brain region specific plots here.
- Figure 4: As already mentioned, I am not sure if it is useful to split the analysis into different brain regions, since also the correlations in the measured different brain areas are very similar.
- Table 2: please report also the interactions and if significant also the detailed post hoc results

---

## Round 0.2 · Minor Revisions

One Reviewer is happy with your revision, the other requires a few minor and final corrections. Please do incorporate these suggestions and I would be happy to accept the paper.

Reviewer 1 ·

Basic reporting

Manuscript entitled: Evidence for a Key Role of the Right Thalamus in Anesthesia-Awakening Regulation in frogs
The authors have carefully addressed all major and minor comments; here please find some additional observations and suggestions.
COMMENTS:
1. I would recommend a more cautious title, like the one in the previous version of the manuscript. In fact, results in this study about single area interactions are not conclusive and remain speculative, thus the title should reflect this aspect.
2. Line 310, I would suggest to write marginally non-significant effect of the factor brain area (p=0.050). The effect is in fact significant when PE is used as variable instead of ApEn.
3. Line 315 “However there was no significant interaction.” I recommend placing this sentence at line 311, before discussing the post-hoc analyses. Something like “, but no significant interaction between factors was observed.”.
4. In paragraph 317-324 post-hoc analyses for the single brain areas are missing and should be reported since the ANOVA revealed a significant main effect of the factor “brain area”.
5. Paragraphs starting from lines 346 and 368, please consider combining them in a single paragraph, since the second one consists of only 3 lines.
6. Line 363 I believe something is missing in the brackets after “r=- […] the latter”.

Experimental design

no comment

Validity of the findings

no comment

Additional comments

no comment

Reviewer 2 ·

Basic reporting

acceptable

Experimental design

acceptable

Validity of the findings

acceptable

Additional comments

no comments

---

## Round 0.3 · accepted · Accept

The revision addresses all the reviewers requests and I am thus happy to accept this paper for publication.